# Intelligent Detection Algorithm for Concrete Bridge Defects Based on SATH–YOLO Model

**DOI:** 10.3390/s25051449

**Published:** 2025-02-27

**Authors:** Lanlin Zou, Ao Liu

**Affiliations:** College of Automotive and Transportation Engineering, Wuhan University of Science and Technology, Wuhan 430081, China; liuonao@126.com

**Keywords:** bridge defect detection, lightweight architecture, StarNet, Adaptive Intra-Feature Interaction, task-dynamic detection, feature fusion

## Abstract

Amid the era of intelligent construction and inspection, traditional object detection models like YOLOv8 struggle in bridge defect detection due to high computational complexity and limited speed. To address this, the lightweight SATH–YOLO model was proposed in this paper. First, the Star Block from StarNet was used to build the STNC2f module, enriching semantic information and improving multi-scale feature fusion while reducing parameters and computation. Second, the SPPF module was replaced with an AIFI module to capture finer-grained local features, improving feature-fusion precision and adaptability in complex scenarios. Lastly, a lightweight TDMDH detection head with shared convolution and dynamic feature selection further reduced computational costs. With the SATH–YOLO model, parameter count, computation, and model size were reduced significantly by 39.9%, 8.6%, and 36.2%, respectively. Meanwhile, the average detection precision was not impacted but improved by 1%, which meets the demands of edge devices and resource-constrained environments.

## 1. Introduction

With the rapid pace of urbanization and the accelerated development of infrastructure, bridges serve as critical nodes in transportation networks, where their safety and reliability directly affect public safety and the seamless operation of economic and social systems. Among existing bridge structures, concrete bridges are one of the most widely utilized types, owing to their superior performance, long service life, and low maintenance costs [1]. However, as bridges age, internal damage and deterioration diminish their load-bearing capacity. Cracks, as a direct manifestation of structural damage, are a primary focus in visual bridge inspections [2]. The presence of cracks accelerates the corrosion of reinforcement steel, significantly shortens the lifespan of bridge structures, and undermines their overall strength and stability. Severe penetrating cracks can result in critical structural damage, posing substantial threats to bridge safety. Consequently, the development of efficient and precise methods for detecting concrete bridge defects has become a pressing issue in the maintenance and management of bridges.

Traditional bridge inspection methods primarily rely on manual inspections and basic physical equipment, which are characterized by low efficiency, high costs, and a dependency on subjective judgment. These limitations make it challenging to detect defects in a timely and accurate manner, especially given the growing demand for routine maintenance of a large number of bridges. Against this backdrop, intelligent detection technologies have gained significant attention in recent years.

The rise of computer vision, particularly the rapid advancement of deep learning techniques, has offered novel solutions for bridge defect detection [3]. For example, Zheng et al. [4] proposed a high-precision concrete crack detection method based on Seg-Net and Bottleneck Architectures. Wang et al. [5] developed a crack detection model using the Inception-ResNet-v2 algorithm, enabling precise segmentation of target areas. Zhang et al. [6] introduced an improved You Only Look Once v3 (YOLOv3)-based method for bridge crack recognition, combining MobileNet with convolutional attention modules to achieve real-time detection of bridge deck cracks. Liao Yanna et al. [7] enhanced the multi-scale prediction module in the YOLOv3 network to leverage shallow semantic features, improving crack detection accuracy. Huang Keyuan et al. [8] utilized YOLOv5 with attention modules to further enhance the accuracy and efficiency of concrete bridge crack recognition. Li Pei et al. [9] proposed an automatic crack detection and classification approach using YOLOv5s with SENet and Coordinate Attention modules. INAM et al. [10] developed a two-stage algorithm that combines YOLOv5 and U-Net. Yang Guojun et al. [2] designed an integrated model for bridge crack recognition and segmentation based on an improved YOLOv7 and SeaFormer, addressing issues such as large model parameters, low detection efficiency, and high latency. Xu Yang et al. [11] applied the Faster R-CNN algorithm to the detection of surface defects of concrete buildings after an earthquake and achieved the recognition and location of four types of defects, including cracks, spalling, exposed steel bars, and buckling steel bars, with an overall average accuracy of 80%. Li Ruoxing et al. [12] proposed an improved Faster R-CNN algorithm to classify and locate concrete defects such as spalling, cracks, and exposed steel bars, with an accuracy of 80.7% for defect classification and 86% for location. Maeda et al. [13] embedded the single-stage SSD algorithm into mobile devices for the detection of concrete pavement defects, successfully detecting a variety of common concrete surface defects, achieving real-time detection of defects, and improving the efficiency of concrete pavement defect detection.

While these methods have achieved notable success in crack detection, their practical application in real-world bridge engineering is often constrained by factors such as limited computational resources, model complexity, and high parameter and computational requirements [14,15]. To address these challenges, this study proposes SATH–YOLO, an improved lightweight model based on YOLOv8. SATH–YOLO aims to facilitate efficient and precise crack detection in practical bridge engineering scenarios.

## 2. SATH–YOLO Algorithm Framework

### 2.1. Overview of the YOLOv8 Algorithm

As depicted in Figure 1, the YOLOv8 algorithm comprises three primary components: the Backbone, the Neck, and the Prediction Head. Compared to YOLOv5, YOLOv8 introduces the C2f module in both the Backbone and Neck, replacing the C3 module. This enhancement improves the model’s lightweight characteristics and facilitates gradient flow information, thereby boosting feature-representation capabilities.

In the Prediction Head, YOLOv8 employs a decoupled head design, separating the classification and detection tasks. This decoupling allows for independent task optimization, resulting in improved average precision and detection stability. Furthermore, YOLOv8 eliminates the traditional anchor-based mechanism, opting instead for an anchor-free approach. This transition simplifies the processes of anchor generation and bounding box refinement, reduces reliance on IoU computations, and significantly decreases memory and computational demands during training.

For positive and negative sample assignment, YOLOv8 introduces the Task-Aligned Assigner mechanism, which dynamically adjusts the ratio of positive to negative samples during training. This feature enhances the model’s adaptability across diverse datasets and tasks.

In terms of loss functions, YOLOv8 employs Binary Cross-Entropy (BCE) loss for classification and integrates Complete IoU (CIoU) loss with Distribution Focal Loss (DFL) for regression. This design aligns seamlessly with the anchor-free paradigm, accelerating the learning of target positional distributions and enabling the model to locate objects more quickly and accurately.

### 2.2. STNC2f Module

The YOLO series models [16,17,18,19] commonly employ a combination of linear projection and nonlinear activation functions. To enhance the feature extraction capabilities of these models, researchers often introduce self-attention mechanism modules based on this combination. These modules construct attention matrices through dot-product operations; however, the computational complexity increases significantly as the feature extraction capability improves.

To address this issue, Microsoft proposed a novel lightweight network named StarNet in 2024 [20], whose architecture is illustrated in Figure 2. StarNet adopts a model design based on star-shaped operations, significantly improving the ability of the model to map inputs to high-dimensional feature spaces, while maintaining low computational complexity. This enhancement ultimately bolsters the model’s feature extraction capacity [21].

The network employs a traditional hierarchical structure, divided into four stages, with the number of channels doubling progressively in each stage at an expansion factor of four. The network achieves down-sampling through convolutional layers and incorporates Star Blocks, which include depthwise convolution, for feature extraction. After the depthwise convolution layers, batch normalization (BN) is utilized instead of layer normalization (LN) to improve computational efficiency.

This design demonstrates exceptional performance and computational efficiency in fields such as natural language processing and computer vision, gradually being applied to lightweight network architectures. The star-shaped operation integrates features from two linear transformations through element-wise multiplication, with the computational formula shown below:(1)$$O_{star}=\left(W_{1}^{T}X\right)*\left(W_{2}^{T}X\right)$$(2)$$W=\left[\begin{array}{l} W\\ B \end{array}\right]$$(3)$$X_{aug}=\left[\begin{array}{l} X\\ 1 \end{array}\right]$$

In the equations: $O_{star}$ represents the result of the star-shaped operation; W denotes the weight matrix of the linear layer; B is the bias of the linear layer; $X_{aug}$ represents the input; and ∗ signifies the star-shaped operation. Specifically, $W_{1}$, $W_{2}\in R^{\left(q+1)(q^{\prime }+1\right)}\;$, $X\in R^{\left(d+1\right)\times n}$.

To simplify the analysis, StarNet focuses on scenarios involving single-output channel transformations and single-element inputs. Based on Equations (1)–(3), it is further defined that $W_{1}$, ${\;W}_{2}$, $X\in R^{\left(q+1\right)\times 1}$, here q is the number of input channels. Thus, the star-shaped operation can be further expressed as:(4)(w1T x)∗(w2T x)=∑m=1q+1w1m xm ∗∑n=1q+1w2n xn =∑m=1q+1 ∑n=1q+1w1m xm w2n xn=α(1,1) x1x1+⋅⋅⋅+α(4,5) x4x5+⋅⋅⋅+α(q+1,q+1) xq+1xq+1q+2q+12
where m, n represents the channel index, and α denotes the coefficient for each term.(5)$$\alpha_{\left(m,n\right)}=\left\{\begin{array}{l} w_{1}^{m}w_{2}^{n}\;m=n,\\ w_{1}^{m}w_{2}^{n}+w_{1}^{n}w_{2}^{m}\;m\neq n. \end{array}\right.$$

By further elaborating on the star-shaped operation described in Equation (1), it can be expanded into a combination of different terms, as shown in Equation (4). Apart from the special terms, each remaining term exhibits a nonlinear relationship with x, indicating that these terms represent independent, implicit dimensions. After computation in a q-dimensional space using the star-shaped operation, the resulting implicit dimensional feature space has a dimensionality of $\frac{\left(q+2)(q+1\right)}{2}\approx {\left(\frac{d}{\sqrt{2}}\right)}^{2}$. This significantly amplifies the feature dimensions without incurring any additional computational overhead within a single layer.

By stacking multiple layers, the star-shaped operation can recursively increase the implicit dimensionality to nearly infinite levels:(6)$$O_{l}=W_{l,1}^{T}O_{l-1}*W_{l,2}^{T}O_{l-1}\;\;\in R^{{\left(\frac{d}{\sqrt{2}}\right)}^{2^{\;l}}}$$
where $O_{l}$ represents the output of the l-th star-shaped operation.

As expressed in Equation (6), using l layers of star-shaped operations enables the construction of an implicit dimensional feature space belonging to $R^{{\left(\frac{d}{\sqrt{2}}\right)}^{2^{l}}}$.

Inspired by the outstanding lightweight design and information extraction capabilities of StarNet, the YOLOv8 baseline model is improved in this work. Given that the overall architecture of StarNet is relatively complex, a direct introduction would result in model redundancy, which could negatively affect the lightweight characteristics of the model. Therefore, this work only introduces the Star Blocks structure, replacing the Bottleneck in the original C2f module in the Backbone and Neck parts with Star Blocks of StarNet, forming the STNC2f module. This modification aims to enhance the network’s feature extraction capabilities and multi-scale information fusion, while reducing both the model’s parameter count and computational complexity. The improved STNC2f module structure is shown in Figure 3.

### 2.3. Intrascale Feature Interaction

The Spatial Pyramid Pooling Fast (SPPF) module is a crucial component in YOLOv8 for enhancing feature-extraction efficiency and performance. By pooling features at different scales, SPPF effectively captures contextual information and improves the spatial invariance of the model. The Attention-based Intrascale Feature Interaction (AIFI) module, introduced in the RT-DETR model [22], is a key element that employs an attention mechanism for intrascale feature interaction, thus improving the efficiency and effectiveness of feature extraction. Its core lies in the application of attention mechanisms between features of the same scale, enhancing the model’s focusing ability and facilitating richer feature fusion. The structure of the AIFI module is illustrated in Figure 4.

The feature fusion process of the AIFI module is shown in Figure 5: First, the 2D S5 feature map is converted into high-dimensional vectors, which are then processed through the AIFI module. The mathematical process involves the output of Multi-Head Self-Attention (MHSA) being added to the input via residual connections and normalized by Layer Normalization to efficiently capture complex relationships within the input sequence. Subsequently, these features are passed through a Feed-Forward Network (FFN) for nonlinear transformation and feature extraction. Finally, the output is normalized again, generating attention scores that encapsulate the critical feature information, which is then returned to 2D form, denoted as F5, for subsequent cross-scale feature fusion. AIFI processes only the S5 feature map to leverage its rich semantic information, reducing computational complexity while preserving detection performance.

Based on this, the improved model in this study replaced the SPPF module with AIFI and added an additional convolutional layer (Conv) before AIFI. This combination helps capture the dependencies between features and improve the efficiency of feature handling and fusion, thereby it is promising in reinforcing the overall detection performance and maintaining good precision for large, medium, and small object detection.

### 2.4. Task Dynamic Mutual Detection Head (TDMDH)

YOLOv8 employs the widely used decoupled head structure in the detection head, which splits the regression and classification tasks into independent branches, each containing two 3 × 3 convolutions and one 1 × 1 convolution. While this design improves detection performance, it significantly increases both the parameter count and computational load. Moreover, YOLOv8 uses a traditional single-scale feature prediction module, which overlooks the impact of features at other scales on detection performance. As a result, YOLOv8 struggles to fully utilize its performance advantages in resource-constrained environments, such as edge devices. To clear this hurdle, we are proposing a customized task-aligned detection head structure, Task Dynamic Mutual Detection Head (TDMDH). Inspired by the concepts of Tood [23], this structure further optimizes model performance through task-aligned label assignment strategies. TDMDH generates joint features by interacting with information from different scales, significantly reducing the model’s parameter count while ensuring that the average precision remains consistent. This makes the model more lightweight and better suited for deployment in resource-constrained environments.

Regarding normalization techniques, Batch Normalization (BN) has proven effective in improving training speed and convergence. However, its strong dependence on batch size can negatively affect performance. Therefore, we replace BN in the convolutional layers with Group Normalization (GN) to overcome the limitations of BN and prevent accuracy degradation due to lightweight operations. The structure is illustrated in Figure 6.

The TDMDH structure is shown in Figure 7. The three feature layers ($P_{3}$, $P_{4}$, $P_{5}$) from the Neck are processed sequentially through two shared convolutional modules (Conv_GN), each with a 3 × 3 kernel. The information sharing between the two convolution layers effectively reduces both the parameter count and computational load. These features are then concatenated using a Concat operation, forming interactive features with combined information. These interactive features are then separated into regression and classification branches via a Task Decomposition module.

In the regression-shared branch, the interactive features generate offsets and masks, which are combined with Deformable Convolutional Networks v2 (DCNv2) [24]. DCNv2 adjusts the convolutional kernel positions and weights more flexibly, especially effective for handling complex geometric transformations such as cracks. Finally, a 1 × 1 convolution predicts the boundary box offsets, and a Scale layer adjusts the output features to accommodate different object scales.

In the classification shared branch, the Dynamic Features Select (DYFS) module performs dynamic feature selection on the interactive features and generates corresponding weights to enhance the recognition of damage targets. Finally, a 1 × 1 convolutional layer predicts the probabilities of each class. The DYFS structure is shown in Figure 8.

### 2.5. SATH–YOLO Detection

This paper proposes a lightweight improved model based on YOLOv8—SATH–YOLO (StarNet AIFI TDMDH-YOLO), whose structure is shown in Figure 9.

Based on the above components, we introduced the Star Block into the Backbone and Neck of YOLOv8, replacing the Bottleneck in the original C2f module with StarNet’s Star Blocks to form the STN-C2f module. This modification achieves the lightweight design of the Backbone and Neck networks, while ensuring detection accuracy. Next, we replaced the SPPF module at the end of YOLOv8’s Backbone with the AIFI (Attention-based Intrascale Feature Interaction) module from the RT-DETR model. This self-attention operation on high-level features not only avoids intrascale interaction of low-level features but also effectively reduces computational redundancy. Finally, the detection head of YOLOv8 is replaced with the TDMDH detection head. Traditional object detection heads typically have separate classification and localization branches, which lack interaction between the two tasks. TDMDH, by extracting features from multiple convolutional layers, learns task interaction characteristics, generates joint features, and significantly reduces model parameter count through shared convolution and dynamic feature selection.

## 3. Experiment-Related Preparation

### 3.1. Experimental Equipment and Hyperparameters

The equipment and some of the hyperparameters used in the experiments are listed in Table 1.

### 3.2. Experimental Dataset

Although the use of public benchmark datasets is crucial for model validation, there is currently a limited availability of public datasets for concrete bridge defects, particularly those that encompass a wide range of defect types and diverse environmental conditions. Therefore, the dataset utilized in this study was initially acquired through field photography of bridge crack defects under various environmental conditions, lighting variations, and shooting angles to ensure diversity and representativeness. The equipment used for image capture was an iPhone 13 Pro Max. To enhance the robustness of the dataset, data augmentation techniques were applied. These included operations such as rotation, scaling, translation, mirror flipping, and the addition of noise to the original images, generating a diverse set of crack image samples that simulate different environmental and operational conditions. This approach ultimately aimed to improve the generalization capability of the model.

After the image data collection was completed, manual annotation was performed using the semi-automatic labeling tool, labelImg. The final dataset comprised 4691 images. Using a dataset partitioning program, the dataset was randomly split into a training set (3801 images), a validation set (678 images), and a test set (212 images).

### 3.3. Evaluation Metrics

To objectively assess the effectiveness of the proposed network model in detecting concrete bridge defects, the following evaluation metrics were selected: precision (P), recall (R), mean average precision (mAP), giga floating point operations per second (GFLOPs), parameter count, and model size. Precision (P) measures the probability of correctly detecting the target, recall (R) determines whether the target can be identified within the complete dataset, and mAP represents the average precision (AP) value across all categories. The closer P, R, and mAP are to 1, and the smaller the computation volume, parameter count, and model size, the higher the model’s recognition accuracy and lightweight performance.

The specific formulas are as follows:(7)$$R=\frac{T_{P}}{T_{P}+F_{N}}$$(8)$$P=\frac{T_{P}}{T_{P}+F_{P}}$$(9)$$F_{AP}=\int_{0}^{1}p\left(r\right)\;\;dr$$(10)$$F_{mAP}=\frac{1}{n}\sum_{i=1}^{n}F_{APi}$$

In Equations (7) and (8), True Positive ($T_{P}$) refers to the number of correctly predicted positive samples; False Positive ($F_{P}$) indicates false positive cases, where the location of the target is detected but the target’s class is misidentified; False Negative ($F_{N}$) refers to the number of undetected targets. Equation (9) defines the mean average precision ($F_{AP}$), which is the average of precision values obtained at recall rates between 0 and 1. In Equation (10), n represents the number of classes in the dataset, and $F_{APi}$ refers to the average precision of the i-th class. The final $F_{mAP}$ is the mean of the average precision (AP) across all classes. For the mAP evaluation in this study, we utilized mAP@50, which represents the average of  AP values when the Intersection over Union (IoU) threshold is set to 0.5.

## 4. Experiments and Discussion

### 4.1. Visual Comparative

To intuitively evaluate the detection performance of SATH–YOLO, a comparison was made with the baseline model, YOLOv8, in terms of crack detection results, as shown in Figure 10. Additionally, Grad-CAM [25] was employed to generate heatmaps that demonstrate the attention of both YOLO models on the crack regions and their respective detection performance. The visual results are presented in Figure 11, where the red and yellow regions in the heatmaps indicate the areas with higher attention from the models.

From Figure 10, the detection boxes of SATH–YOLO exhibited higher boundary precision and more stable box positioning across various crack shapes and lengths. Columns a and c showed that SATH–YOLO captured the full extent of the cracks with reduced background false detections and incomplete detections.

By comparing the visual results of different samples in Figure 11, the following observations can be made:In rows a and b, the heatmap of YOLOv8 shows a slightly diffused attention range, failing to fully align with the edges of the cracks, resulting in less precise detection box boundaries. In contrast, the heatmap of SATH–YOLO better covers the entire crack region, demonstrating higher detection stability.In row c, SATH–YOLO clearly focuses on the core area of the crack, while YOLOv8’s response is unevenly distributed, diminishing detection reliability.In row d, the YOLOv8 heatmap exhibits significant background interference, with the detected area failing to clearly highlight the crack path. In contrast, the SATH–YOLO heatmap effectively reduces interference and accurately identifies crack features.

In summary, compared to YOLOv8, SATH–YOLO demonstrates superior visual attention concentration and boundary recognition precision in crack detection. The positioning capability of its detection boxes and its ability to focus on the target region in the heatmap highlight SATH–YOLO’s enhanced robustness and sensitivity in handling complex backgrounds, thereby providing higher reliability and accuracy for detection tasks.

### 4.2. Ablation Study

To validate the effectiveness of each improvement in the proposed model, an ablation study was conducted for each modified module. The experimental process strictly controlled model parameters to ensure consistency. The evaluation results of the ablation study are shown in Table 2.

As observed from Table 2, the introduction of each individual improvement leads to a certain degree of performance enhancement. Compared to the baseline model, the introduction of the STNC2f module reduces the model’s parameter count by 16.9%, the computation volume and model size by 14.8%, with only a 0.4% decrease in the mean average precision (mAP@50). Replacing the SPPF module in the backbone with the AIFI module enables AIFI to selectively perform self-attention on high-level features containing rich semantic information, preventing internal scale interactions of low-level features, thereby capturing the relationships between conceptual entities in the image. This approach aids in more efficient detection and recognition of objects by subsequent modules. As a result, the parameter count is reduced by 3.7%, computation volume by 1.2%, and model size by 2.2%, demonstrating that AIFI effectively reduces computational redundancy.

Introducing the Task-Dynamic Multi-Task Head into the detection head leads to information sharing and task decoupling mechanisms between the convolutional layers, regression, and classification branches. This enables the learning of joint features from multi-scale information interactions and dynamically selects relevant features to generate corresponding weights. This results in a significant reduction in both parameter count and model size, while also slightly improving mAP. Specifically, the parameter count decreases by 26.9%, model size by 24.7%, and mAP@50 increases by 0.3%, with computation volume increasing by just 0.6%.

When the STNC2f, AIFI, and TDMDH modules are combined, SATH–YOLO achieves significant improvements across all metrics compared to the baseline model. The parameter count decreases by 37.9%, computation volume by 8.6%, model size by 36.2%, and mAP@50 increases by 1%. These experimental results suggest that SATH–YOLO dynamically selects features and acquires rich semantic information, thereby improving performance while achieving model lightweighting.

### 4.3. Comparative Experiments

To further validate the performance of the SATH–YOLO algorithm in detecting concrete bridge crack defects, it was compared with other high-performance models in the YOLO series, including YOLOv3-tiny, YOLOv5n, YOLOv6, YOLOv8n, YOLOv9t, YOLOv10n, and YOLOv11n. The experimental environment and parameter configurations strictly followed the settings outlined in Section 2.1, with the experimental results shown in Table 2. The mAP@50 curves for all algorithms after training for 300 epochs are presented in Figure 12.

The experimental results indicate that YOLOv3-tiny, YOLOv6, and YOLOv8n have relatively high parameter counts, computation volumes, and model sizes, with subpar mAP@50 performance. While YOLOv5n and YOLOv9t perform well in terms of lightweight design, their parameter counts and model sizes are still higher than those of SATH–YOLO, and their detection mAP@50 is 0.8% and 1% lower, respectively. YOLOv10n shows similar mAP@50 performance to SATH–YOLO but has a 121.4% and 145.5% higher parameter count and model size, respectively. YOLOv11n, although having a computational advantage with a 14.9% reduction in computation volume compared to SATH–YOLO, has lower mAP@50 and a parameter count and model size 138% and 137.6% larger than SATH–YOLO’s. See Table 3.

As shown in Figure 12, SATH–YOLO’s mAP@50 curve remains consistently superior throughout the 300-epoch training process, with minimal fluctuations in the stable phase, indicating higher detection accuracy and stability. SATH–YOLO not only matches the convergence speed of other advanced models but also exhibits significant advantages in final detection performance. This demonstrates that the improvements in feature extraction and detection robustness effectively enhance the model’s overall performance.

Due to resource limitations, this study was unable to conduct testing on edge devices. However, future experiments will focus on deploying the StarNet SATH–YOLO model on platforms such as the NVIDIA Jetson Nano and Raspberry Pi. These tests will evaluate the model’s real-time performance and practicality in resource-constrained environments. The planned experiments will include assessing inference speed, memory usage, and power consumption, providing insights into the model’s suitability for edge deployment in practical applications.

### 4.4. Statistical Analysis and Significance

To assess the statistical significance of the performance improvement of the proposed SATH–YOLO model over the YOLOv8 baseline, an independent two-sample *t*-test was performed. The null hypothesis (H_0_) posits no significant difference between the models, while the alternative hypothesis (H_1_) asserts a significant difference in performance. A significance level of 0.05 was adopted. The *t*-test was conducted based on the accuracy values obtained from five independent training runs for each model.

The t-statistic is computed as follows:(11)$$t=\frac{\bar{X_{1}}-\bar{X_{2}}}{\sqrt{\frac{s_{1}^{2}}{n_{1}}+\frac{s_{2}^{2}}{n_{2}}}}$$
where $\bar{X_{1}}$, $\bar{X_{2}}$ are the mean accuracies for YOLOv8 and SATH–YOLO, respectively, $s_{1}^{2}$, $s_{2\;}^{2}$ are the sample variances, and $n_{1}$, $n_{2}$ are the sample sizes.

The average accuracy of YOLOv8 was 97.2% (SD = 0.20%), while SATH–YOLO achieved 98.2% (SD = 0.16%). The t-test resulted in a t-statistic of 10.87 and a *p*-value of 4.54 ×${\;10}^{-6}$, indicating a statistically significant performance difference (*p* < 0.05). See Table 4.

The results demonstrate that the performance improvement observed in SATH–YOLO is statistically significant, as the *p*-value of 4.54 ×${\;10}^{-6}$ is well below the 0.05 threshold. This confirms that the observed performance difference is not due to random chance, providing strong statistical evidence for the superiority of the proposed model.

These findings suggest that SATH–YOLO not only improves accuracy but also does so with statistical significance, further reinforcing its effectiveness and efficiency, particularly for deployment in resource-constrained environments.

## 5. Conclusions

With the widespread application of deep learning in practical scenarios, particularly in edge devices or resource-constrained environments, model lightweighting has become key to enhancing the applicability of models [26,27,28]. In order to address the issue that current mainstream object detection algorithms, such as YOLOv8, which struggle to fully exploit their performance advantages in resource-limited and edge-device environments, the concrete bridge disease detection model SATH–YOLO was proposed to improve the accuracy and efficiency of concrete bridge defect detection.

With the SATH–YOLO model, the C2f module in the Backbone and Neck networks was replaced with the STN-C2f module composed of Star Blocks from the lightweight StarNet network; the traditional SPPF feature fusion methods was replaced with AIFI module focusing on high-level semantic feature interaction; the lightweight TDMDH was utilized to minimize the parameter count and computation.

Compared to the baseline model YOLOv8, the SATH–YOLO model showed significant improvement in parameter count, computation volume, and model size, with no impact on precision. The experimental results from this study showed that SATH–YOLO is a promising model that allows the bridge detection to run efficiently with limited computational resources and low-power devices, accelerating inference speed, reducing power consumption and hardware costs, and enabling rapid deployment. Therefore, the proposed concrete bridge disease detection method based on SATH–YOLO has broad application prospects, especially in large-scale bridge monitoring tasks, where it can effectively reduce manual inspection workloads and improve the comprehensiveness and accuracy of defect identification. Despite the promising results of the SATH–YOLO model for crack detection in bridges, this study has several limitations. First, the model’s evaluation was limited to crack detection, and its generalizability to other types of structural damage remains untested. Additionally, the dataset used in this study may not fully represent real-world variability, and the lack of real-world testing on edge devices limits the practical assessment of the model’s deployment. Furthermore, the model’s stability and generalization ability in extreme environments need further optimization. Future research should focus on expanding the model to detect other types of damage, improving the dataset with more diverse and representative data, and testing the model on actual edge devices such as the Jetson Nano or Raspberry Pi to validate its feasibility for real-time deployment in resource-constrained environments. Moreover, exploring integrations with other detection technologies, such as drone inspections and laser scanning, could enhance the model’s robustness and advance the automation and intelligence of bridge defect detection. Overall, the proposed algorithm is of significant theoretical importance and practical feasibility, with strong potential for widespread application.

## Figures and Tables

**Figure 1 sensors-25-01449-f001:**
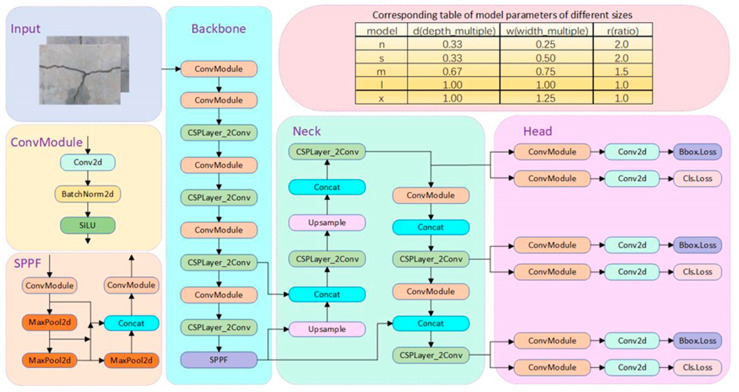
Structure diagram of the YOLOv8 algorithm.

**Figure 2 sensors-25-01449-f002:**
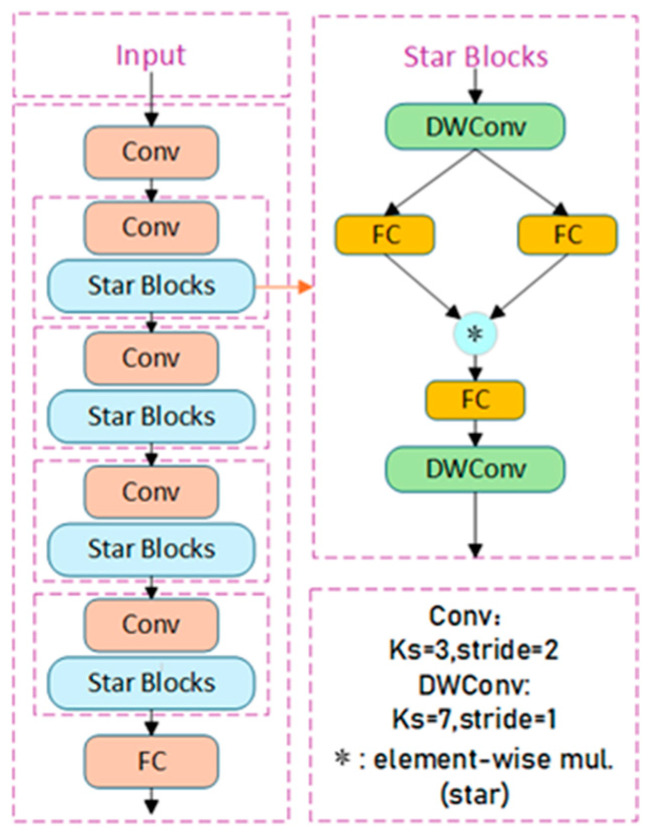
StarNet Architecture Principles.

**Figure 3 sensors-25-01449-f003:**
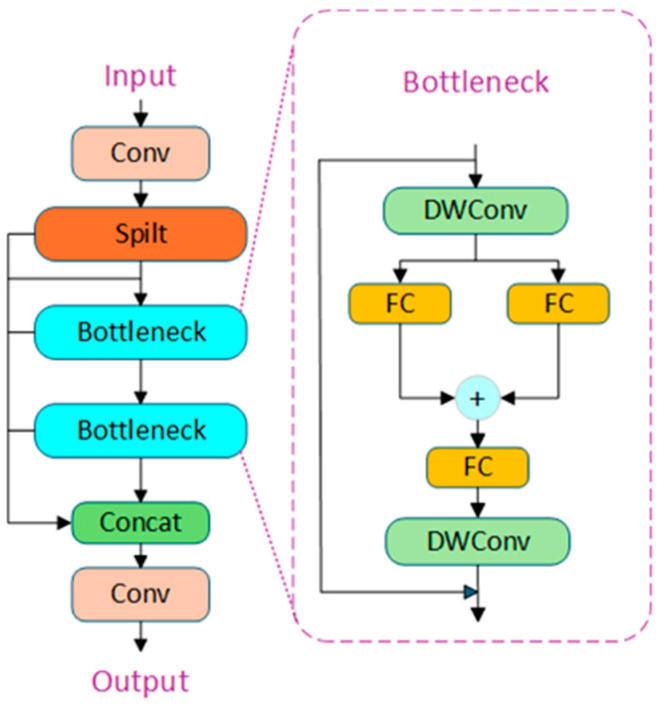
STNC2f module structure.

**Figure 4 sensors-25-01449-f004:**
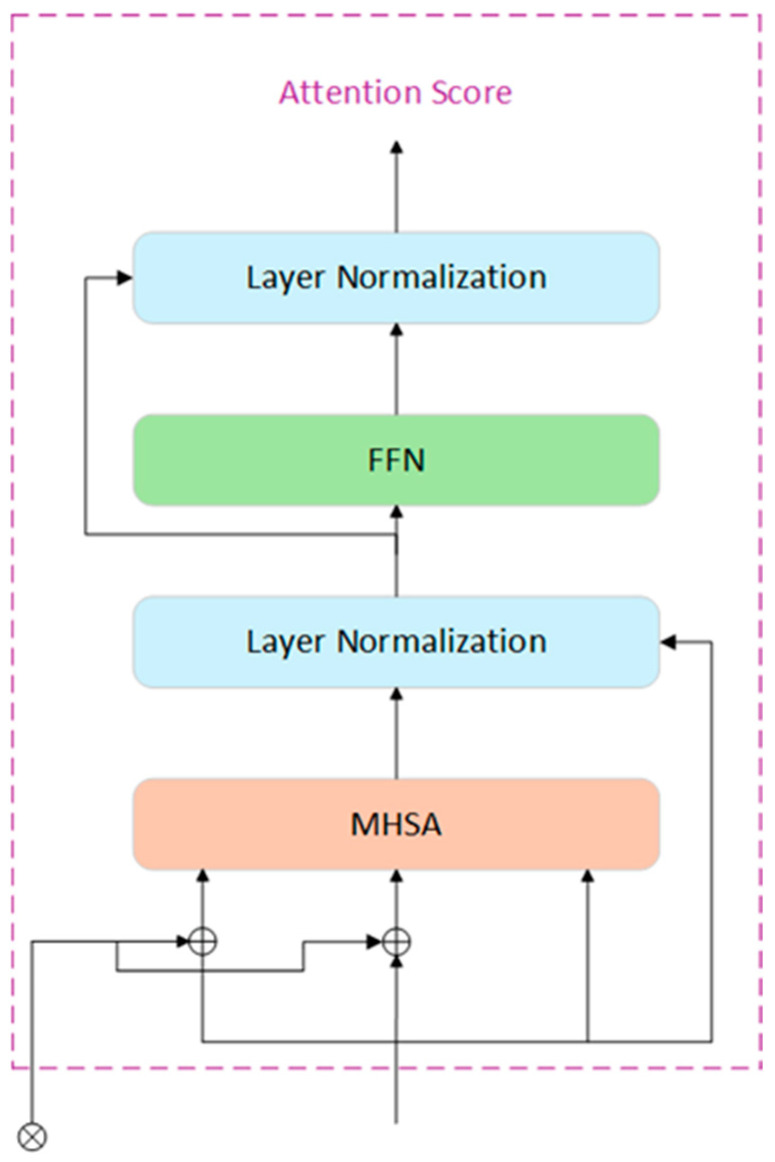
AIFI module structure.

**Figure 5 sensors-25-01449-f005:**
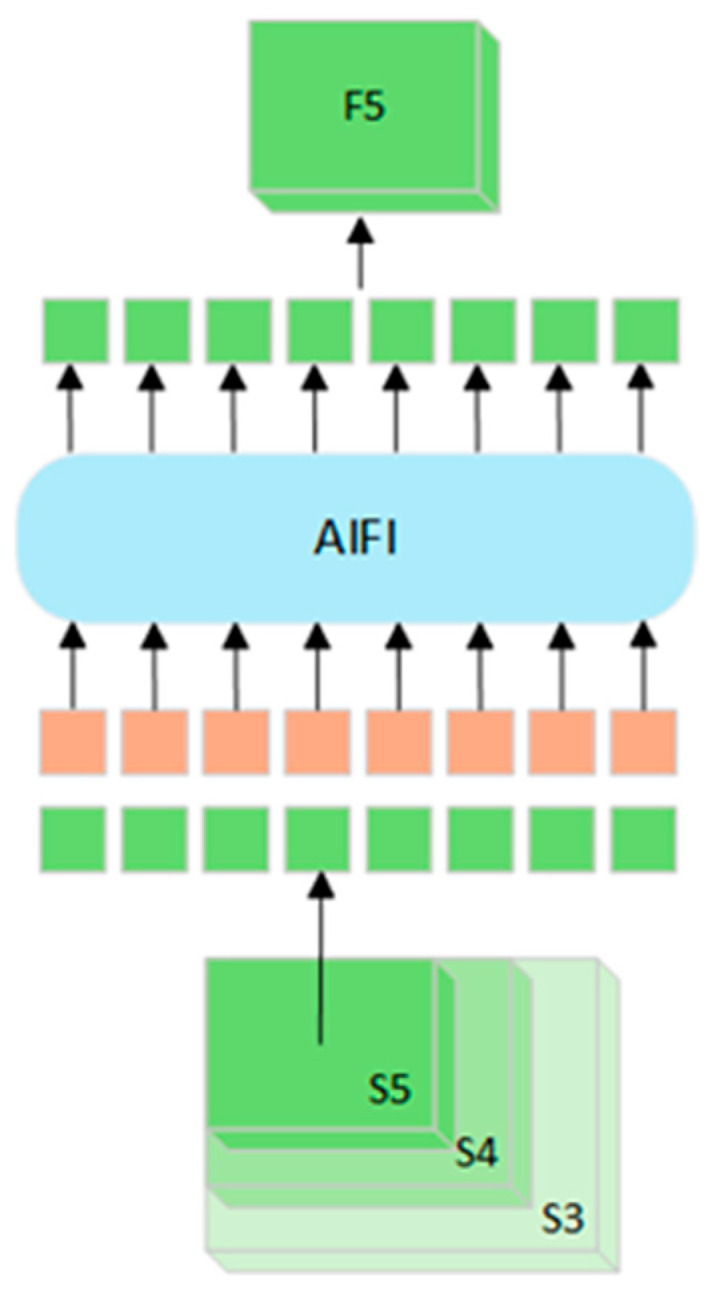
The feature fusion process of the AIFI module.

**Figure 6 sensors-25-01449-f006:**
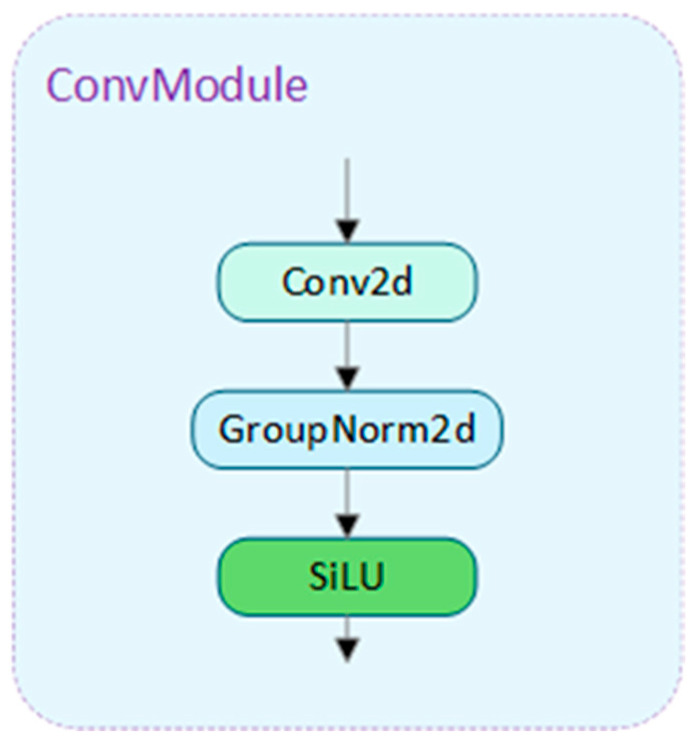
Conv_GN module structure.

**Figure 7 sensors-25-01449-f007:**
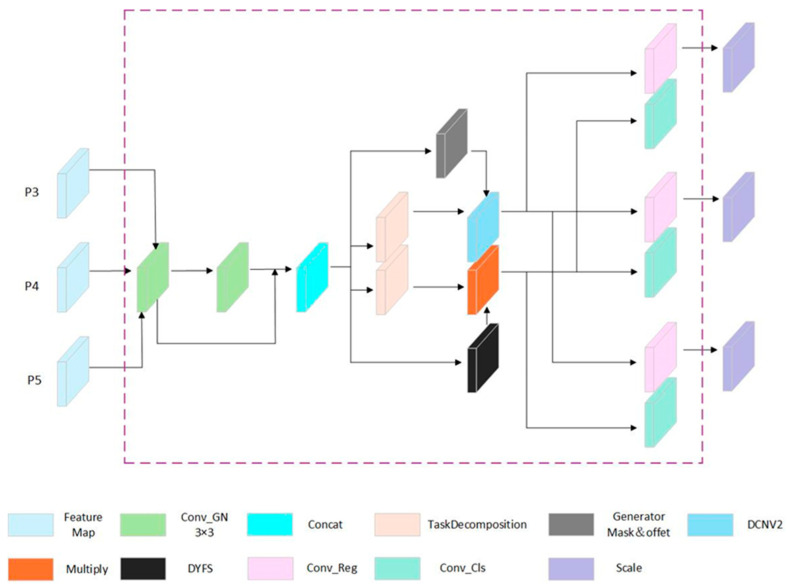
TDMDH structure diagram.

**Figure 8 sensors-25-01449-f008:**
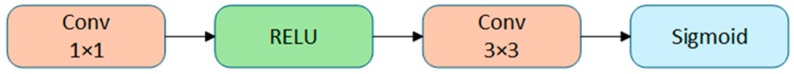
DYFS structure diagram.

**Figure 9 sensors-25-01449-f009:**
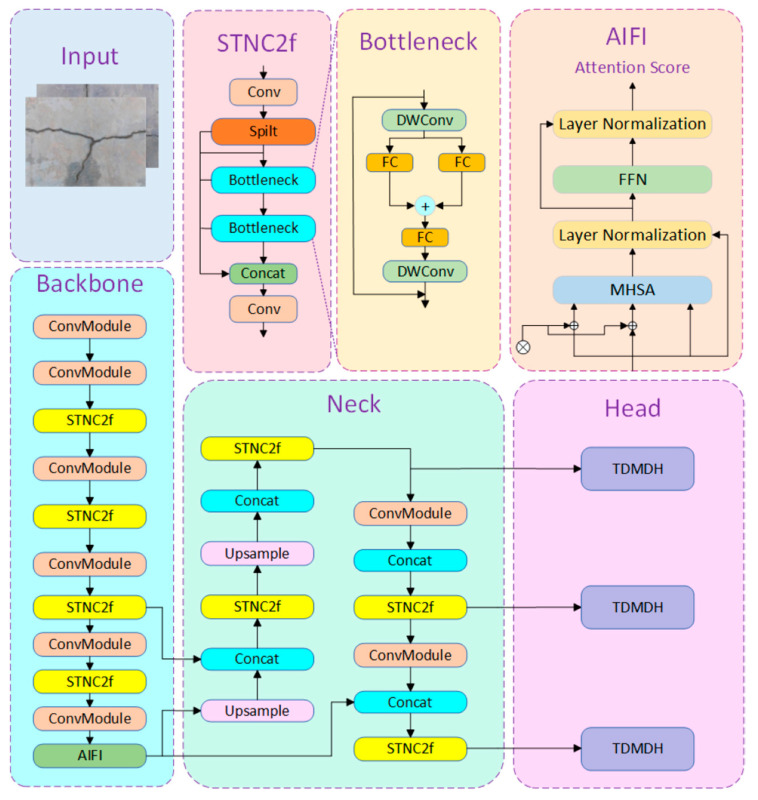
SATH–YOLO structure diagram.

**Figure 10 sensors-25-01449-f010:**
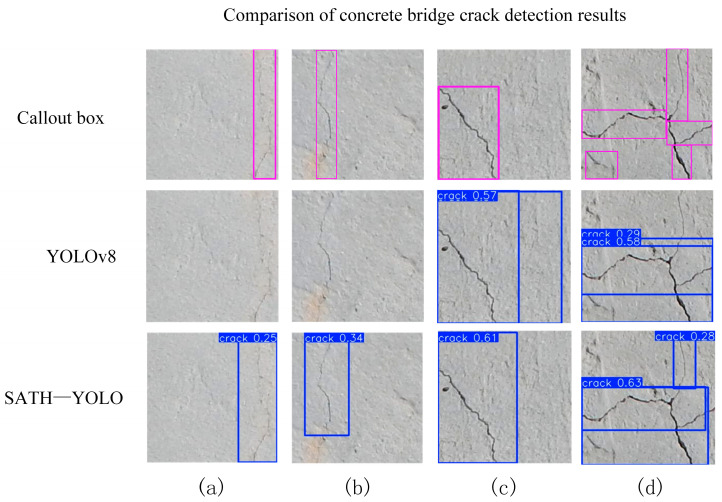
Comparison of concrete bridge crack detection results. (**a**–**d**) are four types of random cracks.

**Figure 11 sensors-25-01449-f011:**
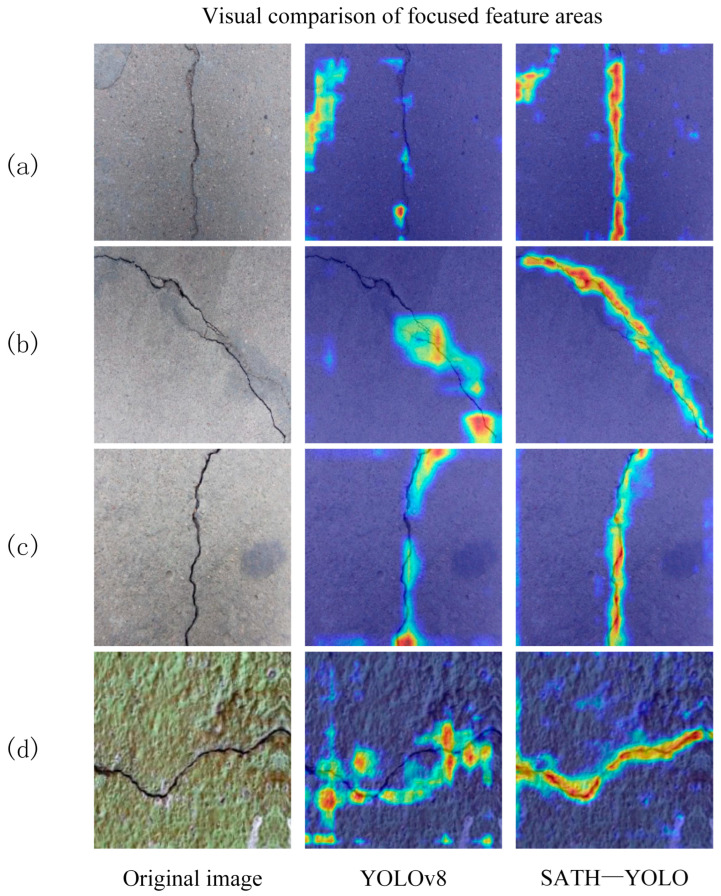
Visual comparison of focused feature areas. (**a**–**d**) are four types of random cracks.

**Figure 12 sensors-25-01449-f012:**
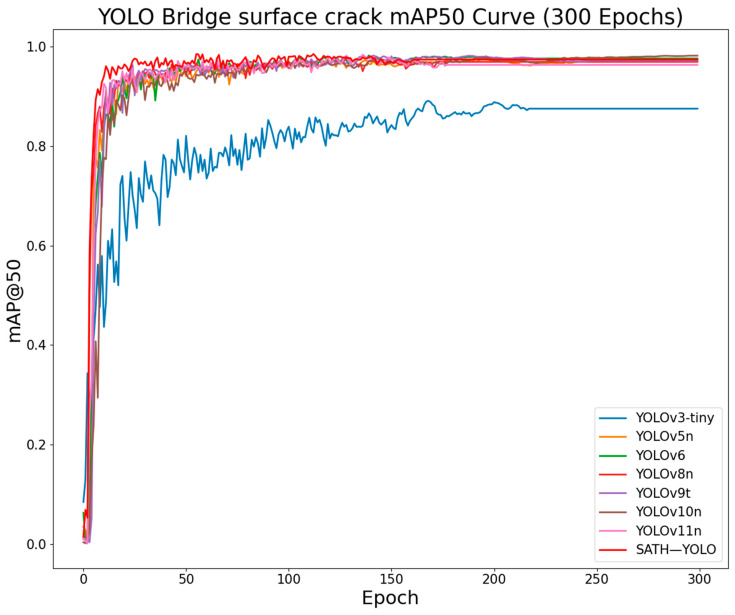
Comparison of mAP@50 curves of YOLO series on bridge crack dataset.

**Table 1 sensors-25-01449-t001:** The equipment and some of the hyperparameters used during the experiments.

Device	Parameter	Hyperparameter	Value
CPU	Intel(R) Core(TM) i7-12700KF	Training Strategy	SGD
RAM	32 GB	Momentum	0.937
Operating System	Windows 10 Pro	Initial Learning Rate	0.01
GPU	NVIDIA GeForce RTX 4060Ti	Final Learning Rate	0.0001
GPU Memory	8GB	Batch Size	16
Programming Tool	Visual Studio Code 1.96.4	Workers	5
Programming Language	Python 3.9.19	Imgsz	640
Deep Learning Framework	PyTorch 2.3.1	Total Epochs	300

**Table 2 sensors-25-01449-t002:** Ablation experiment table.

STNC2f	AIFI	TDMDH	Params/M	Flops/G	Precision/%	Recall/%	mAP@50/%	Model Size/MB
—	—	—	3.01	8.1	96.5	92.4	97.2	5.96
√	—	—	2.50↓	6.9↓	97.5↑	92.8↑	96.8	5.08↓
√	√	—	2.90↓	8.0↓	95.9	94.4↑	96.9	5.83↓
—	—	√	2.20↓	8.6	95.2	93.3↑	97.5↑	4.49↓
√	√	—	2.53↓	7.1↓	97.0↑	94.0↑	97.9↑	5.09↓
√	—	√	1.85↓	7.1↓	96.4	94.7↑	97.8↑	3.75↓
—	√	√	2.19↓	8.5	97.8↑	91.7	97.6↑	4.40↓
√	√	√	1.87↓	7.4↓	98.8↑	94.1↑	98.2↑	3.80↓

“√” indicates that this module is introduced; “—” indicates that this module is not introduced; ↑ and ↓ both represent the parts that have been improved compared to the baseline model, where ↑ represents an increase and ↓ represents a decrease.

**Table 3 sensors-25-01449-t003:** Comparative Experiment Table.

Model	Params/M	Flops/G	Precision/%	Recall/%	mAP@50/%	Model Size/MB
YOLOv3-tiny	12.13	18.9	79.9	82.2	88.9	23.2
YOLOv5n	2.50	7.1↓	98.2	93.4	97.4	5.05
YOLOv6	4.23	11.8	97.6	90.3	97.3	8.30
YOLOv8n	3.01	8.1	96.5	92.4	97.2	5.96
YOLOv9t	1.97	7.6	95.7	94.6↑	97.9	4.45
YOLOv10n	2.27	6.5↓	95.1	94.3↑	98.2	5.53
YOLOv11n	2.58	6.3↓	97.1	95.3↑	97.7	5.23
SATH–YOLO	1.87	7.4	98.8	94.1	98.2	3.80

↑ and ↓ both represent the improvement compared to SATH–YOLO, where ↑ represents an increase and ↓ represents a decrease.

**Table 4 sensors-25-01449-t004:** Performance Comparison and Statistical Significance of YOLOv8 and SATH–YOLO.

Model	Accuracy (%)	Standard Deviation (%)	t-Statistic	*p*-Value	Statistical Significance
YOLOv8	97.2	0.20			
SATH—YOLO	98.2	0.16	10.87	4.54 × $10^{-6}$	Significant (*p* < 0.05)

## Data Availability

The data presented in this study are not available upon request from the corresponding author due to the involvement of personal privacy in some of the monitoring footage.

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
