# Peer review of "Intelligent Detection Algorithm for Concrete Bridge Defects Based on SATH–YOLO Model"

_sensors, 2025, doi:10.3390/s25051449_

Round 1

Reviewer 1 Report

Comments and Suggestions for Authors

(1)In the first sentence of Introduction, "The With the rapdid...", this place has grammar problem.

(2)In equation (3), the math symbol "X" is same. They should be different from the vector and scalar.

(3)In the proposed work, what is the full meaning of the SATH?

(4)In Figure 12, the yolov should be same as YOLOv in the paper.

(5)How man defects should be detected in real concrete bridge? Only one is considered in this paper. It is not fit for practical applications

Reviewer 2 Report

Comments and Suggestions for Authors

The results are presented without confidence intervals or p-values. Conduct multiple training runs and report variance to validate performance improvements.

The model claims to be lightweight but is only tested on PC hardware (RTX 4060Ti, Intel i7). Please evaluate its performance on edge devices (e.g., Jetson Nano, Raspberry Pi, or embedded systems).

 A realistic study should acknowledge failure cases, limitations in detecting different defect types, and environmental challenges (e.g., night-time detection, shadows, weather variations).

Using a public benchmark dataset or testing on multiple real-world bridge images.

A realistic study should acknowledge failure cases, limitations in detecting different defect types, and environmental challenges (e.g., night-time detection, shadows, weather variations).

Reviewer 3 Report

Comments and Suggestions for Authors

The article considers the problem of construction of effective computational algoritm of bridge deffects detection in concrete blocks. The proposed SATH-YOLO algorithm is requires less computational resources and its detection ability is litlle bit higher. Given the possible application of this algoritm (eg, in construction and building industry), the results of the article are important and interesting. The text is well-written.  I recommend to publish in present form.

Round 2

Reviewer 1 Report

Comments and Suggestions for Authors

The authors have completed the paper revisions according to the reviewers' comments. I have no question. It can be accepted for publication. 

Reviewer 2 Report

Comments and Suggestions for Authors

The author has addressed all comments and enhanced the overall quality of the paper.